# Evaluation of the adverse events following immunization surveillance system, Ghana, 2019

**Eunice Baiden Laryea[1], Joseph Asamoah Frimpong[1]***, **Charles Lwanga Noora[1‡],
John Tengey[1‡], Delia Bandoh[1‡], George Sabblah[2‡], Donne Ameme[1☯], Ernest Kenu[1☯],
Kwame Amponsa-Achiano[3☯]**

**1** Ghana Field Epidemiology and Laboratory Training Program, Department of Applied Epidemiology and
Disease Control, School of Public Health, University of Ghana, Accra, Ghana, **2** Department of
Pharmacovigilance, Food and Drugs Authority, Accra, Ghana, **3** The Expanded Program on Immunization,
Ghana Health Service, Accra, Ghana

☯ These authors contributed equally to this work.
‡ CLN, JT, DB and GS also contributed equally to this work.
* asamoah.frimpong@gmail.com

SOUTH AFRICA

**Data Availability Statement:** All relevant data are
within the paper and its Supporting Information
files.

## Abstract

### Background

With over 80% of children worldwide vaccinated, concerns about vaccine safety continues
to be a public health issue. Ghana's Adverse Events Following Immunization surveillance
started in 1978 with the objective to promptly detect and manage AEFI cases either real or
perceived. Periodic evaluation of the surveillance system is critical for optimal performance;
hence we evaluated the system to assess its attributes, usefulness and system's perfor-
mance in meeting its objectives.

### Methods

A case of AEFI was defined as any untoward medical event occurring within 28 days after
vaccination and may not necessarily have causal relationship with the vaccine use. We
reviewed surveillance data and procedures for the period 2014 to 2018 and interviewed key
stakeholders. Adapting the CDC's Updated Guidelines for Evaluating Public Health Surveil-
lance Systems, we assessed the system's attributes and usefulness. We performed sum-
mary descriptive statistics on quantitative data and directed content analysis on information
gathered from interviews.

### Results

In all, 2,282 AEFI cases including 476 (21%) serious cases (life threatening events) were
reported for the period. The highest case detection rates of 61.45 AEFIs per 100,000 surviv-
ing infants was recorded in 2018. Reporting forms were modified to accommodate new indi-
cators without any disruption in the function of the system. At the national level,
completeness of 100 randomly sampled reporting forms (100%) and was higher than the
region (27%) but timeliness (50%) was lower than the region (83%). All (16/16) Community

**Funding:** The author(s) received no specific funding for this work.

**Competing interests:** The authors have declared that no competing interests exist.

Health Nurses interviewed indicated "fear of being victimized" as the reason for underreporting, nonetheless, the system was useful as it made them cautious when vaccinating children to prevent reactions. Data on AEFI surveillance was also useful in guiding training needs and provision of vaccination logistics.

## Conclusion

The AEFI surveillance system is useful at all levels but partially meeting its objective due to underreporting. We recommend training and supportive supervision to improve timeliness of reporting, data completeness and acceptability.

## Introduction

Vaccination is an important public health tool that prevents about 2 to 3 million deaths worldwide yearly [1]. Generally, vaccines are safe and effective and undergo extensive safety monitoring prior to its usage in immunization programs. However, no vaccine is entirely without risk and Adverse Events Following Immunization (AEFI) may occur [1] which can be detected through Post market surveillance.

Adverse Events Following Immunization (AEFI) is any untoward medical occurrence that follows immunization and does not necessarily have a causal relationship with usage of the vaccine [2]. The adverse event may be any unfavorable or unintended sign, abnormal laboratory finding, symptom or disease. AEFIs may range from mild reactions such as fever or rash to severe reactions such as convulsions, coma and even death [3]. Based on the outcome of the event, the World Health Organization (WHO) has classified AEFIs as Serious and Non-serious which is used for regulatory classification. AEFIs are further classified based on the cause of the event. These include, vaccine product-related, vaccine quality defect-related, immunization error-related, immunization anxiety-related, coincidental and unknown events [4]. Hence, AEFIs may be true adverse reactions (ARs) to the vaccine, the vaccination process or may be unrelated.

Though over 80% of children worldwide are vaccinated, vaccine hesitancy continues to be a public health issue [5]. Surveillance on immunization safety is hence crucial to build trust and to reassure the public that AEFIs are being monitored and actions are being taken to reduce risks. This will reduce vaccine-hesitancy and sustain efforts by immunization programs [6]. Reporting rates of AEFIs are usually low, especially in developing countries [7]. This is due to over reliance on the routine system of reporting which is subject to underreporting and poor pharmacovigilance infrastructure in developing countries [8]. To address the issue of underreporting the WHO therefore instituted the Global Vaccine Safety Initiative, which set out indicators primarily for monitoring case reporting [9]. A case reporting target of 10 AEFI per 100,000 surviving infants per year was set in the Global Vaccine Action Plan (GVAP) to monitor performance of AEFI surveillance systems [10].

In 2015, the average reporting rate was 549 AEFI per 100,000 surviving infants globally. The number of countries that reported rates greater than 10 per 100,000 surviving infants also increased from 8 (4%) in 2000 to 81(42%) in 2015 [11]. Meanwhile, only half (51%) of the countries in the WHO African Region, reported AEFI cases in 2015, and the region recorded one of the lowest case reporting rates of 74 AEFIs per 100,000 surviving infants [11]. Studies done in some African countries have also reported incidence of AEFI ranging from 13 to 34% [3, 12].

In Ghana, routine reporting of AEFI began with the initiation of Expanded Program on Immunization in 1978 and also by the Food and Drugs Authority (FDA) in 2001 upon establishment of the National Pharmacovigilance Centre [4]. The surveillance system was established with an overall aim of promptly detecting and managing AEFI, real or perceived. This would contribute to the credibility of immunization program and prevent inappropriate responses to reports of AEFI that could lead to crises in the absence of a surveillance system.

We evaluated Ghana's AEFI surveillance system to assess its attributes, its usefulness and to determine whether the system is meeting its objectives.

## Methods

### Evaluation design

The evaluation was a cross-sectional study that involved collection of data through reviewing of surveillance data, observation of surveillance procedures and interviewing of key stakeholders. The evaluation was conducted in January 2019 and covered the period from 2014 to 2018.

### Study area

The evaluation was done in Ghana using Greater Accra Region as the focal region and Ga East Municipality, as the focal district. Interviews were done among selected stakeholders from the district, regional, and national levels, and surveillance records were sampled at all these level for the evaluation. In 2018, Ghana's population of children under 5 years of age targeted for routine and supplementary childhood immunization was 5,922,867 with 912,194 fully immunized. The Greater Accra Region also had an under 5 years population of 579,805 with 136,150 fully immunized, while Ga East District which is one of the 26 districts in Greater Accra Region had under 5 years population of 22,431 with 3,455 children fully immunized. The district has 16 health facilities, one health centre, one polyclinic and one Quasi Government. It also has 14 private health centres and 15 Community-based Health Planning and Services (CHPS) zones of which one has a compound. There is one health care worker to every 132 children under five who offer immunization and related services. Health worker training on AEFI is done prior to every supplemental immunization activity.

### Data collection tools and methods

We adapted the Centre for Disease Control and Prevention Updated Guidelines for Evaluating Public Health Surveillance Systems [13]. This tool provides indicators to measure the qualities of a surveillance system. Using these indicators, we developed an interview guide and checklist to suit the objectives of the evaluation and describe system's operations. Thirty key informants from health facilities (19) district (6), regional (3) and national (2) levels were purposively selected and interviewed based on their role in AEFI surveillance. Questions were asked regarding operations of the surveillance system. We observed surveillance processes and reviewed data (both paper and electronic-based) from 2014 to 2018 at all levels. Data was collected on the number and type AEFI, timeliness and completeness in reporting among others. The sources of data were; a) case-based forms, b) weekly and monthly Integrated Disease Surveillance c) Response (IDSR) reports, d) DHIMS 2 and e) Technical Advisory Committee for Vaccines and Biological Products (TAC-VBP) reports.

A case of AEFI was defined as any untoward medical event occurring within 28 days after vaccination which does not necessarily have causal relationship with the vaccine use [14]. The adverse reaction was classified as Serious or Non-serious or Clustered [15].

**Serious AEFI.**    An AEFI was considered serious if; It is life threatening, results in inpatient hospitalization or prolongation of existing hospitalization, results in persistent significant disability/incapacity, results in death, or results in a birth defect/congenital anomaly.

**Non-serious AEFI.**    These are events that do not meet the definition of a serious AEFI and usually include mild to moderate symptoms that are temporarily lived. Though common, these must be carefully monitored as they may signal a potentially larger problem with the vaccine or immunization as well as have negative impact on the public acceptance of the vaccine.

**AEFI clusters.**    Two or more AEFIs related in time and/or place or vaccine administered.

### Assessment of performance of the AEFI surveillance system in meeting its objectives

We reviewed the objectives of the surveillance system to substantiate its fulfilment or otherwise. Its objectives are to; detect, correct and prevent immunization error-related AEFIs, identify problems with vaccine lots or brands leading to vaccine reactions, prevent false blame arising from coincidental adverse events, maintain confidence by properly responding to community concerns, generate new hypotheses about vaccine reactions and estimate rates of occurrence of AEFIs. This was assessed through records review and interviews. Summary descriptive statistics, was used to determine number of AEFI cases detected and proportion investigated to determine its cause. Directed content analysis was used to summarize information on how AEFIs were managed at the facilities, ability of the investigation process to refute false claims of vaccine-related reactions and use of data for signal event detection and corrective action such as withdrawal of an implicated vaccine.

### Assessment of usefulness of the AEFI surveillance system

We interviewed stakeholders on; the system's effect on policy decisions and use of data for action, its contribution to the detection and prevention of AEFIs, and its effect on clinical practices. We also assessed ability of the system to achieve its set objectives through records review and interviews. Directed content analysis was used to summarize information gathered.

### Assessment of attributes of the AEFI surveillance system

We described and analyzed each attribute of the system as shown in **Table 1**.

### Ethical considerations

Permission was sought from the Ghana Health Service, Greater Accra Regional Health Directorate and Ga East Municipal Health Directorate. Permission was also sought from Food and Drugs Authority and Expanded Program on Immunization for the use of data. Informed written consent was sought from interviewees without any form of duress. Data obtained was put into Microsoft excel which was password protected. This evaluation was done as part of routine assessment by the Ghana Health Service and Ghana Field Epidemiology and Laboratory Training Program (GFELTP).

## Results

### Background characteristics of study participants

Thirty health staff were interviewed of which 21 (70%) were females. The mean (SD) age and years of service among respondents was 38.7 (4.1) years and 13.1(2.3) years respectively. At the facility level, sixteen (53.3%) community health nurses, two (6.7%) medical officers, one (3.3%) pharmacist were interviewed while at the district level, four (13%) disease control/

**Table 1. Assessment of attributes of the AEFI surveillance system.**

| Attributes | Data collection | Data analysis |
|---|---|---|
| Simplicity | Interviews | Directed content analysis to summarize health workers' knowledge on case definition, ease of; confirming a case, completing the case-based forms, flow of information through the levels of the system and case investigation. |
| Acceptability | Interviews | Directed content analysis to describe evidence on stakeholders' participation in the surveillance system at all levels |
| | Records review | |
| Sensitivity | Records review | Summary descriptive statistics to determine proportion of regions/districts (or the years) that achieved the GVAP case detection target of 10 AEFI/100,000 surviving infants/year |
| Positive Predictive Value | Records review | The number of AEFIs related to the vaccine or the vaccination process out of all the serious AEFIs for which causality was assessed.<br><br>$*PPV = \frac{Number\ of\ AEFIs\ vaccine-related}{Number\ of\ AEFIs\ that\ causality\ was\ assessed} \times 100$ |
| Timeliness | Records review | Summary descriptive statistics to determine proportion of AEFIs that are reported to the next level on time and feedbacks received on time as well as turnaround time for AEFI investigation. |
| | Interviews | |
| Stability | Interviews | Directed content analysis to summarize availability and sustainability of funds, logistics and human resources needed to run the system as well as level of integration of the system into the IDSR. |
| Data quality | Records review | Summary descriptive statistics to determine proportion of case-based forms, that are completely filled. Directed content analysis to describe data validation. |
| Representativeness | Records review | Directed content analysis to describe reported events by age, sex, time of report, and place (sub-district/district/region) of reporting |
| Flexibility | Records review | Directed content analysis to summarize how the system had adapted to changing information needs and operation conditions with little increase in cost, time and funds. |
| | Interviews | |

surveillance officers, two (6.7%) Health Information Officers were interviewed. At the regional level three (10%) key informants; the Regional DCO, DSO and EPI focal person were interviewed and at the national level two (6.7%) key informants; from EPI Program Manager and Head of Pharmacovigilance of FDA were interviewed.

Out of the 30 respondents, 26(87%) understood the purpose of the AEFI surveillance system and 28(93%) could define AEFI. All respondents knew at least two presenting signs/symptoms of AEFI, while 24(80%) knew the correct timelines for submission of forms. Six (20%) did not know how to fill a reporting form all of which were community health nurses.

## Operations of the surveillance system

The operation of the AEFI surveillance system is a collaborative effort between the Expanded Program on Immunization of the Ghana Health Service and the Food and Drugs Authority (FDA). The system involves collection and collation of routine data using the health structures of the Ghana Health Service. Case reporting is passive where caregivers/vaccinees report adverse events to health facilities. The health facilities record reported events using a standard case reporting form and submits the report to the District Health Directorate. The District focal person enters the data on the forms into District Health Information Management System II (DHIMS 2) and transmits the form to the National level through an intervening Regional Focal Point. Reports must be backed by case reporting forms and validated before entry into the DHIMS. Data are further aggregated at the regional and national levels. Also, some notifications are sent directly from the community or the health facility to the FDA through an electronic reporting system. Once such notification is received, focal points at the FDA complete the standard AEFI form (Fig 1).

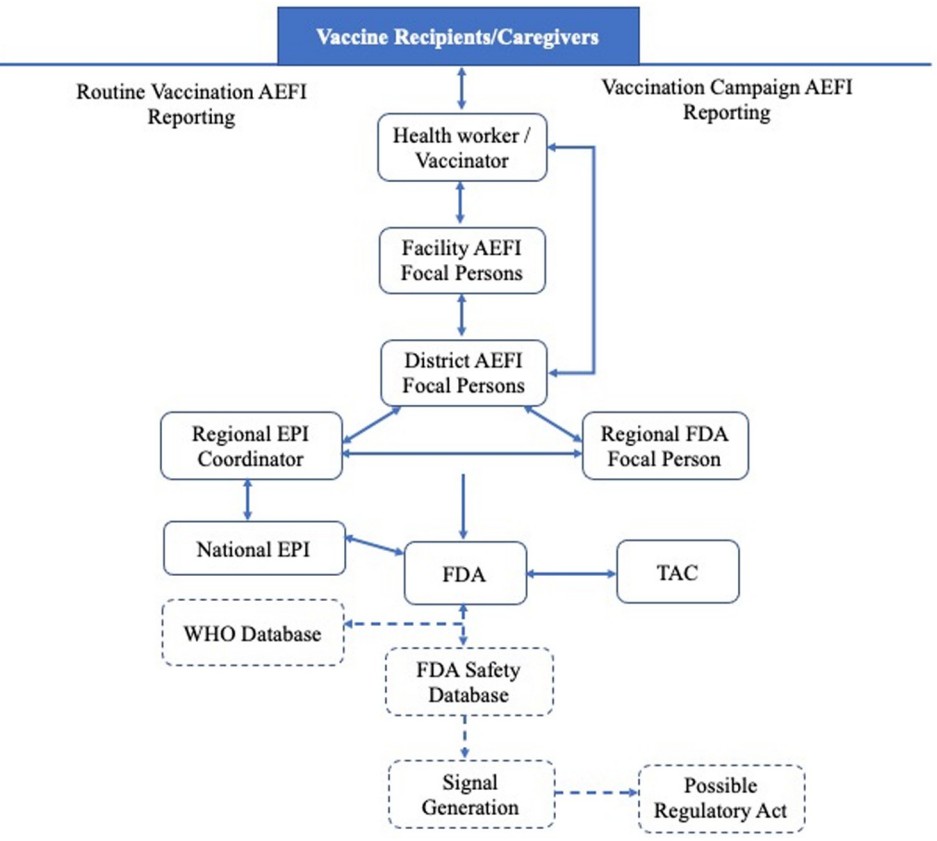

**Fig 1. Information pathway, AEFI surveillance system, Ghana. Source:** Guidelines for Surveillance of AEFI in Ghana [15].

Investigation of an AEFI is initiated within 24 to 48 hours of receipt of report at the lowest level with the capacity to carry out that investigation. Causality assessment is done on serious AEFI cases and AEFI clusters by a Technical Advisory Committee for Vaccines and Biological Products (TAC-VBP) constituted by the FDA. Meetings of the Committee are held at interval of at most two months. Feedback back on receipt of report is given within 48 hours to the reporting facility/level. However, feedback on serious AEFI is given bimonthly when the TAC-VBP meets for causality assessment.

Sustainability of the system depends on the availability of adequate resources at each level of the surveillance system. The Gavi Alliance is a major financier of the immunization Program and AEFI surveillance in Ghana. Other Partners include WHO and UNICEF. Tools for AEFI surveillance include AEFI reporting form, investigation form, guidelines for completing AEFI reporting form, electronic line listing form, clinical laboratory form and the guideline for AEFI Epidemiological investigation.

## Assessment of performance of the surveillance system in meeting objectives

**Objective 1: Detect, correct and prevent immunization error-related AEFIs.** The surveillance system detected a total of 2,282 AEFI cases including 21% (476/2282) serious cases during the 5-year period (2014–2018) of which 15% (72/476) were immunization error related. All (16/16) CHNs interviewed acknowledged that the presence of AEFI surveillance makes them more cautions when vaccinating in order to prevent immunization error-related AEFIs.

Health workers had also taken measures to prevent immunization error-related AEFIs which include stopping student nurses on attachment from administering vaccines.

**Objective 2: Identify problems with vaccine lots or brands leading to vaccine reactions.** The AEFI case-based form captures the lot/batch number of vaccines that the vaccinee took prior to the reaction. This aids to the identify any problems vaccine lot during investigation. About 97% (463/2282) of serious cases were investigated using the lot/batch number of vaccines.

**Objective 3: Prevent false blame arising from coincidental adverse events.** By investigating deaths that followed vaccination and performing causality assessments the system has been able to prevent misinformation speculating that the vaccine attributed to the death.

**Objective 4: Maintain confidence by properly responding to parent/community concerns.** The awareness campaigns and health education given on AEFIs during immunization sessions increased public confidence. Also, all reported AEFIs were managed at no cost to the affected individuals as this cost was absorbed by the Expanded Program on Immunization. This made caregivers more comfortable to vaccinate their children and report any events.

**Objective 5: Generate new hypotheses about vaccine reactions and estimate rates of occurrence of AEFIs.** The system was able to monitor trends of AEFI cases and detect unusual trends in number of cases, however there was no documentation on incidence of AEFIs in Ghana.

## Assessment of usefulness of the AEFI surveillance system

The reporting form captures demographic features of the vaccinee as well the characteristics of the vaccine taken. This informs the age groups at risk and the antigens causing the most adverse reaction for public health action to be taken. The EPI coordinators at the regional and national level revealed during interviews that data on AEFI surveillance was useful in guiding training needs and provision of vaccination logistics. Lastly, the system led to change of clinical practices as all CHNs indicated that the system made them extra careful during injection in order to prevent abscess and ensured that vaccines are stored properly and used within required time.

## Assessment of attributes of the AEFI surveillance system

**Simplicity.** Sequential relay of information in the system was clear and well structured. Ninety-three percent (28/30) of stakeholders interviewed knew and understood the AEFI case definition. Identifying a case involved observing the individual and could be captured through easily detectable symptoms requiring basic tools like a thermometer. The case reporting form was a one-page document, with 25 core variables which during interviews, 80% (24/30) stakeholders said it takes less than 15 mins to fill. On the other hand, serious cases required laboratory confirmation and causality assessment which had longer turn-around time especially for a coroner's case. The case investigation form is a four-page document with many variables that required some level of expertise to fill. Additionally, the perfect characterization of AEFI is elusive due to absence of pathognomonic signs or available laboratory tests that would allow confirmation of diagnosis.

**Acceptability.** Stakeholder participation at the national level was high, proven by periodic meetings to review data, analyze and discuss ways to improve the system. Support from WHO, UNICEF and GAVI also demonstrated high acceptance of the system by partners. Stakeholders interviewed at regional level revealed that the system is essential and easy to run. Meanwhile, at the district, all 16 community health nurses interviewed addressed concerns of being victimized for reported events **(Fig 2).**

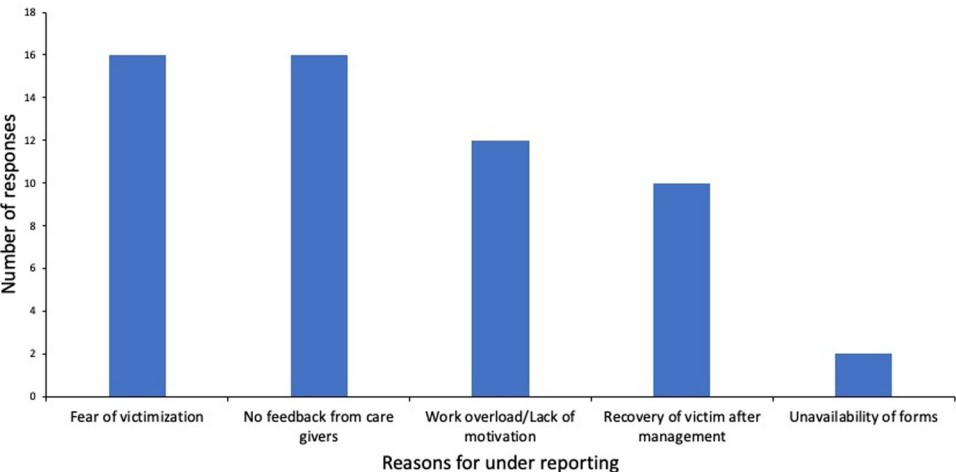

**Fig 2. Reasons for underreporting AEFIs.**

**Sensitivity.** The case definition was sensitive, capturing any untoward medical event after vaccination whether caused by the immunization or not. At the national level, the GVAP case detection target of 10 AEFI/100,000 surviving infants /year was met only in 2017 and 2018 with 35.69 and 61.45 AEFIs per 100,000 surviving infants respectively (**Table 2**). In 2018, 80% (8/10) of regions in Ghana achieved this target based on routine vaccination reports (**Table 3**). At the district level, the annual case detection rate for 2014, 2017 and 2018 was 4.54, 5.83 and 4.54 AEFIs per 10,000 surviving infants respectively. No case was detected in 2015 and 2016 (**Table 4**).

**Positive Predictive Value (PPV).** PPV is the proportion of AEFIs that are actually caused by the vaccine. This was assessed at the national level since causality assessment is only done at this level. For 2014 and 2015, PPV could not be calculated due unavailability of data, however in 2016, 2017 and 2018 the country recorded PPVs of 66.7%, 50% and 63.6% respectively.

**Timeliness.** Seven case reporting forms were found at the district of which two were submitted to the region on time. At the region, 83(83%) of 100 forms randomly selected were submitted by districts on time. At the national level, 50% (50/100) of reports were timely submitted (within 28 days from onset of symptom) in 2017 and 2018 and 80% of forms were timely submitted in 2016. Upon interviewing surveillance staff at the district and sub-district levels, 88% (22/25) stated that they did not get feedback when they sent reports to higher levels.

**Stability.** The AEFI surveillance system is run on the Ghana Health Service (GHS) structures and health workers report data as part of their routine work requiring no extra financial benefit. Cased-based forms were readily available either at the facility or online to be downloaded and printed when needed. The AEFI surveillance guidelines was available at all levels.

**Table 2. Annual performance on AEFI reporting in Ghana, 2014–2018.**

| Year | AEFI cases | #Surviving infants | AEFIs/100,000 surviving infants/year |
|------|-----------|--------------------|--------------------------------------|
| 2014 | 10 | 1024577 | 0.98 |
| 2015 | 20 | 1048147 | 1.91 |
| 2016 | 41 | 1071855 | 3.83 |
| 2017 | 391 | 1095677 | 35.69 |
| 2018 | 688 | 1119618 | 61.45 |

**Table 3. Regional performance on AEFI reporting in Ghana, 2018.**

| Region | AEFI cases | Total pop. (million) | #Surviving infants | AEFI/100,000 surviving infants/year |
|---|---|---|---|---|
| Ashanti | 49 | 5.66 | 215808 | 22 |
| Brong Ahafo | 47 | 2.78 | 105640 | 44 |
| Central | 128 | 2.52 | 95760 | 13 |
| Eastern | 34 | 3.17 | 120460 | 28 |
| Northern | 75 | 2.99 | 113620 | 66 |
| Greater Accra | 169 | 4.83 | 183540 | 92 |
| Upper West | 23 | 0.82 | 31160 | 7 |
| Upper East | 80 | 1.24 | 47120 | 16 |
| Volta | 69 | 2.54 | 96520 | 7 |
| Western | 15 | 3.02 | 114760 | 13 |

All levels also had surveillance focal persons for EPI who served as the AEFI focal person. Surveillance data was stored electronically at all levels in the DHIMS 2 which can be assessed once authorized. This makes it easy to integrate the data into other systems. However, GAVI is the sole financier of immunization and vaccine safety in the country.

**Data quality.** All case investigation forms randomly sampled at the national level had all the 25 core variables completely filled. At the region, only 27(27%) of the 100 forms sampled were completely filled while at the district none of the seven forms available was completely filled. Data imputed into the DHMS were inconsistent with data available at each level of the system. For instance, Ga East recorded 92 AEFI cases in the DHMS in 2018, which proved to be a data entry error.

**Representativeness.** Case reports were from both males and females, however, distribution of cases was mainly among children under five, excluding pregnant women and travelers who took tetanus and yellow fever vaccines respectively. All regions submitted case reports to national level for all years assessed. However, reporting rates among districts to the region was low as only 3 out of 26 districts in Greater Accra submitted AEFI case reports in 2014 and 2015, while 5 districts submitted in 2016 and 2017. At the district level, case reports were mainly from health facilities in 2 out of 5 subdistricts for all years assessed, but included both private and public facilities.

**Flexibility.** The monthly vaccination report and case-based form were modified to accommodate new indicators without any disruption in the function of the system. The AEFI case definition was also altered from a definition that implicated the vaccine use to a definition that the AEFI may or may not be caused by the vaccine. It required a day training for the disease control and surveillance officers at the district levels and on-job coaching for the stakeholder at the facility level. Dissemination of reports is made flexible by inclusion of phone calls and through WhatsApp platforms for instances where paper report submission will delay response. However, the case-based form must be forwarded within 48 hours or be filled by the

**Table 4. Annual performance on AEFI reporting in Ga East Municipality, 2014–2018.**

| Year | AEFI cases | #Surviving infants/year | AEFI/10,000 surviving infants/year |
|---|---|---|---|
| 2014 | 3 | 6610 | 4.54 |
| 2015 | 0 | 6571 | 0.00 |
| 2016 | 0 | 6688 | 0.00 |
| 2017 | 4 | 6858 | 5.83 |
| 2018 | 9 | 7010 | 12.83 |

recipient so as not to compromise the completeness of reports. The reporting form also had a space for additional information to be added.

## Discussion

The surveillance system was found to be functional and well-structured due to a collaborative effort among the Ghana Health Service, the FDA and Partners. Health workers had high knowledge of the system which could be attributed to their longer years of work experience in service (mean years in health service was 13.1). Yet high knowledge among health workers did not reflect reporting rates. This is because the system is heavily reliant on voluntary reporting by caregivers and most of these caregivers were not reporting to the facilities. This may be because caregivers were not educated enough to know that they have to report these events, as found in a study by Constantine et al. [3]. However, Chimusoro et al., had contrary findings where though caregivers had good knowledge on the surveillance system, they still did not report [16]. It could also be because caregivers may feel coming back to the health facility to report is burdensome. To curb this, an active, participant-centered system can be adapted as an adjunct to traditional passive AEFI surveillance. This is an innovation that allows caregivers use communication technologies to report an AEFI to a facility or a central body such as the FDA and has proven to increase reporting rates in other countries [7, 17].

Another reason for low reporting rates was the fear of victimization among the vaccinators. Though most of these health workers knew the purpose of the surveillance system, they still perceived that reporting AEFI will question their work proficiency. This makes the system less acceptable to the vaccinators. Zvanaka et al. had similar findings where health workers had high knowledge on AEFIs yet refused to report due to fear of victimization and fear of exposing work incompetency [18]. This shows that knowledge and perception of health workers towards a surveillance system has great influence on its success [19], hence training must be geared towards debunking this perception.

We found the AEFI case definition to be sensitive as it removes the need to ascertain causality. This also confers a level of simplicity on the surveillance system. However, efforts made to make the system simple and sensitive, did not reflect on case reporting rates especially for 2015 and 2016. High reporting rates in 2018 however could be attributed to the frequent sensitization and training on AEFI during a Yellow fever and Measles-Rubella campaigns held in that year. This goes to prove that routine reporting of AEFI will receive such boost if given similar attention as during campaigns.

Case reporting was simple but case investigation was cumbersome, partly due to the lack of pathognomonic signs and laboratory tests for perfect characterization of an AEFI [20]. As a result of these complexities in the investigation process, assessment of causality is limited to serious AEFIs [4]. Thus, for AEFIs classified as non-serious, its cause may not be identified for corrective actions to be taken.

Representativeness of the system was threatened as the system did not adequately capture AEFIs on Tetanus and yellow fever vaccines received by the pregnant women and travelers respectively. Thus, the ante-natal clinic and the port health clinics must be engaged to report AEFIs. Also, the region had many silent districts which may mean these districts are not sensitized enough to report. A study found representative of the AEFI surveillance system in Kwekwe District in Zimbabwe to be threatened by failure of private facilities to report. This was attributed to low knowledge among health workers in the private sector [16]. This contradicts with our findings where both private and public facilities reported because, immunization services provided at private facilities are mostly carried out by government staff from the various subdistricts as part of their routine work.

Errors in reporting into the DHIMS and inconsistencies in the number of case reports observed at all levels suggest a weak data validation system, Meanwhile, accurate reporting is crucial in informing decision making.

The system was stable with both human and material resources available for operation. However, if GAVI withdraws its funding, the system may collapse. Hence the GHS/EPI and FDA must institute strategies to take over funding of the system if support from GAVI is halted.

The system's flexibility in the use of phone and WhatsApp platforms for reporting reduces the delays in reporting. Nonetheless, human negligence could lead to failure to fill a form and hence gaps in reporting.

The AEFI surveillance system is very useful based on indicators assessed. However, lack of timely feedback from higher levels can eclipse its usefulness. Surveillance staff could be discouraged from sending case-based forms to higher levels, which could impede the use of data for decision making. In Kwekwe District in Zimbabwe, 71.7% of health workers in a study stated that they did not receive feedback from the upper levels and hence were discouraged to forward reports to these levels [16]. Zvanaka et al. also indicated that though the majority of health workers reported the system to be useful, its usefulness is not evidenced by reported feedbacks especially at the district and subdistricts [18]. The usefulness of the system can therefore be improved if national and regional levels give timely and precise feedback on AEFI reports.

Unlike drugs, vaccines are given to seemingly healthy individuals and in vast numbers especially among infants. Thus, issues arising from vaccination are less tolerated by the general public. With this, a more robust surveillance system has to be achieved to maintain public confidence in immunization services to sustain efforts by stakeholders.

This evaluation had a few limitations. There was the possibility of recall bias during interviews as study was conducted in 2019, but required investigation from year 2014. As such, records had to be reviewed to confirm some of the facts obtained during the interviews. Also, most interview questions were asked again among other stakeholders to review the validity of the previous information collected. Again, the researcher relied on AEFI case-based reports and secondary data which could be subject to reporting and data entry errors.

## Conclusion

The AEFI surveillance system is well structured with standardized data collection tools. The system is useful in estimating magnitude of morbidity of AEFIs and improving clinical practices. The system was flexible and stable, with a sensitive case definition however, reporting rates were low due to underreporting. With this the system was partially meeting its objective to promptly detect and manage AEFIs. Acceptability and simplicity were fair, while representativeness, data quality and timeliness were inadequate. We recommended training of vaccinators on AEFI reporting by the AEFI focal persons to counter the fear of being victimized for reported events. We also recommend periodic supportive supervision by Health Management Teams to reporting facilities to ensure improvement system's attributes and use of data for action.

- The West African Health Organisation (WAHO)

- The Ghana Field Epidemiology and Laboratory Training Programme

- The Disease Surveillance Department of the Ghana Health Service.

- The Food and Drugs Authority, Ghana.

- The Expanded Program on Immunization, Ghana.

- Dr. Basil Benduri Kaburi

- Dr Selorm Botwe (Ga East Municipal Director of Health Services)

- Mr. Bright Amoako (Health Information Officer, Ga East Municipal)

## Supporting information

**S1 Dataset. Manuscript dataset.**
(XLSX)

## Acknowledgments

The authors would like to acknowledge the following persons and institutions;

## Author Contributions

**Conceptualization:** Eunice Baiden Laryea, Kwame Amponsa-Achiano.

**Data curation:** Eunice Baiden Laryea, George Sabblah.

**Formal analysis:** Eunice Baiden Laryea, Donne Ameme.

**Investigation:** Eunice Baiden Laryea.

**Methodology:** Eunice Baiden Laryea, Joseph Asamoah Frimpong, Charles Lwanga Noora.

**Supervision:** Joseph Asamoah Frimpong, Charles Lwanga Noora, John Tengey, Delia Bandoh, Donne Ameme, Ernest Kenu, Kwame Amponsa-Achiano.

**Visualization:** Delia Bandoh.

**Writing – original draft:** Eunice Baiden Laryea, Joseph Asamoah Frimpong.

**Writing – review & editing:** Eunice Baiden Laryea, Joseph Asamoah Frimpong, Charles Lwanga Noora, John Tengey, Delia Bandoh, George Sabblah, Donne Ameme, Ernest Kenu, Kwame Amponsa-Achiano.

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
