## [Decision Letter · Decision Letter 0]

19 Aug 2021

PONE-D-21-21870

Evaluation of the Adverse Events Following Immunization Surveillance System, Ghana, 2019

PLOS ONE

Dear Dr. Frimpong,

Thank you for submitting your manuscript to PLOS ONE. After careful consideration, we feel that it has merit but does not fully meet PLOS ONE’s publication criteria as it currently stands. Therefore, we invite you to submit a revised version of the manuscript that addresses the points raised during the review process.

The authors present an evaluation of Ghana’s AEFI surveillance system (i.e. does the system meet the programme’s objectives) using an adaptation of a CDC tool. Vaccine safety is an important area of research and underserved in sub-Saharan Africa and the study has the potential to add to the knowledge base. Unfortunately, in its current form, the manuscript does not meet the criteria for publication.

The overall methodology is very unclear, although some study procedures are described. At present it is difficult to relate the methods to the objectives and results.

Reviewer 2 has offered a very thorough summary with constructive suggestions.

Please also review my more detailed comments in the attached document.

We look forward to receiving your revised manuscript.

Kind regards,

Emma K. Kalk

Academic Editor

PLOS ONE

Journal Requirements:

4. Please upload a copy of Figure 1 and 2, to which you refer in your text on page 6 and 12. If the figure is no longer to be included as part of the submission please remove all reference to it within the text.

Reviewers' comments:

Reviewer's Responses to Questions

**Comments to the Author**

1. Is the manuscript technically sound, and do the data support the conclusions?

Reviewer #1: No

Reviewer #2: Partly

2. Has the statistical analysis been performed appropriately and rigorously? 

Reviewer #1: No

Reviewer #2: N/A

3. Have the authors made all data underlying the findings in their manuscript fully available?

Reviewer #1: No

Reviewer #2: Yes

4. Is the manuscript presented in an intelligible fashion and written in standard English?

Reviewer #1: No

Reviewer #2: No

5. Review Comments to the Author

Reviewer #1: The present manuscript (Evaluation of the Adverse Events Following Immunization Surveillance System, Ghana, 2019) seeks to evaluate Ghana’s AEFI surveillance system. Although the subject regarding AEFI in children is certainly of general interest, the manuscript is not uniform and easy to read, and has several English inaccuracies. Given the low quality of the methods and the poor originality of the findings, the manuscript in its current version is not acceptable for publication.

Reviewer #2: This study is a good assessment of the AEFI reporting system in Ghana and is a valuable contribution to the world of vaccine safety surveillance and system strengthening. A good deal of information has been gathered. However, it is a little difficult to connect the methodology to the results. There would be benefit in revising the layout so that objective and measurable are clearer.

Specific commets:

Abstract:

Line 40: The three updates within last decade appears not to be included in the main body of text

Line 43: ‘All (16/16) Community Health Nurses interviewed indicated “fear of being victimized”’ doesn’t connect to the second part of the sentence; needs to be made more clearer

Introduction:

Line 54: “Vaccine safe and effective; however adverse events following immunization”: this is contradictory; although I agree vaccines are generally safe; perhaps add the word, Generally vaccines are safe and effective;

Also it would be beneficial to add something that vaccines undergo extensive safety testing during clinical trials but rare events can be detected through post marketing surveillance

Line 60 ; serious and non-serious should not be confused with severity; based on the outcome of the event (e.g. hospitalization, death); seriousness is not based on severity; this sentence should be corrected.

Line 66: vaccine hesitancy is a public health issue; I would suggest replacing the term “concerns” with vaccine hesitancy:

67: There should be an explanation on why surveillance on immunization is essential; I.e.: to build trust; to reassure the public that AEFIs are being monitored and actions to reduce risks are being taken

Line 69:AEFI to AEFIs

Line 69: Under reporting is a well-known with routine reporting systems

Line70: the term routine is preferable to passive

Line 70: HICs also rely on routine systems; under reporting is a general limitation of passive/routine systems; but is there any other reason that LMICs have additional underreporting?

Line 70 -72: perhaps rephrase to emphasize the link: underreporting was addressed via the GVSI. One of the targets of… was set.

Materials and methods

Line 93: is there a reference for CDC guidelines for evaluating ?

Line 94: how were these guidelines adapted? What were the quantitative and qualitative data that was collected?

Line 98: Study area paragraph: The evaluation was made through interviewing stakeholders in this region? And reviewing reports in this district? Perhaps clarify what resources in this area was used to do the evaluation.

Line 125: I don’t think the term detect can be applied here; perhaps just state that caregivers/ vaccinees report to health facilities. How do they report? Using a reporting form?

Line 126: What is meant by manage reports? They receive and code them? And they transfer the reports?

Line 153: title says data collection and analysis, but no analysis was described: perhaps expand on the analysis or remove analysis from title.

Also please make data sources clearer :by stating five different sources of data was collected data included: a) interviews .. b) case-based forms; c) IDSR reports; d) DHIMS2; e) TAC.VBP reports

Line 159: explanation of what TAC-VBP reports are?

Line 165: explain what is meant by hypothesis generation; is this signal detection?;

Line 199; prefer to use the word understood rather than knew

Line 205 section on performance of the surveillance system in meeting objectives: objective were; 1) detect, correct and prevent immunization error-related AEFIs; 2)

identify problems with vaccine lots or brands leading to vaccine reactions, 3) prevent false

blame arising from coincidental adverse events, 4) maintain confidence by properly responding

to community concerns, 5) generate new hypotheses about vaccine reactions and estimate rates

of occurrence of AEFIs.

Use subtitles to indicate the results of assessment of each of these objectives;

Line 208: would it be better to say 463/476 (97%) of all serious cases were investigated?

Line 209: suggest to rephrase replace the word exonerate; By investigating deaths that followed vaccination and performing causality assessments the system has been able to prevent misinformation speculating that the vaccine attributed to the deaths? Also link tis to the objective of prevent false blame arising from coincidental adverse events

Line 211: link lot/batch number to objective: identify problems with vaccine lots or brands leading to vaccine reactions

Line 212: I would replace the word victim; perhaps individual;

Line 212: who paid for the management of the AEFIs?

Line 214: it is difficult to get an incidence rate with spontaneous reporting as there is always under reporting. Even if exposure information is known, the actual number of AEFIs collected through spontaneous reporting is an under estimation of incidence. Active surveillance methods are needed to estimate the incidence

Line 212-213: does this refer to signal ( hypothesis generated)? This needs to be made more clearer

Line 215: how was the usefulness assessed? Was this through interviews? Where did this information come from?

Line 223-224: case definition of AEDI reports?,

Line 224: please do not use the word victim; an individual is suspected to have AEFI; causality is not determined; the word victim gives a negative connotation, when most of the time benefits of the vaccine outweigh the risk

Line 225-230: was 15 minutes to fill a report and investigations into serious cases considered to be simple?

Line 280-281: was this a human error or a problem with the tools?

Line 308: in Ghana is this the case, do caregivers have to go back to the facility to report? Or are there tools in place to allow patient/caregiver reporting?

Line 344: full stop (.) after system

Line 349: does Whatsapp platforms compromise the completeness of reports?

Another limitation is recall bias in interviews; as study was conducted in 2019 , investigating from year 2014.

6. PLOS authors have the option to publish the peer review history of their article (what does this mean?). If published, this will include your full peer review and any attached files.

Reviewer #1: No

Reviewer #2: No

---

## [Author Response · Author response to Decision Letter 0]

15 Oct 2021

The Academic Editor

PLOS ONE

Dear Sir

Evaluation of the Adverse Events Following Immunization Surveillance System, Ghana, 2019 (PONE-D-21-21870)

Thank you for the review and providing comments to improve our manuscript. The table below are answers or responses to specific comments made by the reviewers.

COMMENT COORECTIONS MADE

ABSTRACT

Line 40: The three updates within last decade appears not to be included in the main body of text

 The monthly vaccination report and case-based form were modified to accommodate new indicators. The AEFI case definition was also altered from a definition that implicated the vaccine use to a definition that the AEFI may or may not be caused by the vaccine. This has been included in lines 493 to 496 at the results section.

‘All (16/16) Community Health Nurses interviewed indicated “fear of being victimized”’ doesn’t connect to the second part of the sentence; needs to be made more clearer

 The sentence has been revised to build the connection between the two phrases to make the meaning of sentence clearer. This can be seen in line 45 to 48.

INTRODUCTION

“Vaccine safe and effective; however adverse events following immunization”: this is contradictory; although I agree vaccines are generally safe; perhaps add the word, Generally vaccines are safe and effective;

 The word “Generally” was introduced to the sentence in line 60. 

Also, it would be beneficial to add something that vaccines undergo extensive safety testing during clinical trials but rare events can be detected through post marketing surveillance

 The sentence was improved by adding that vaccines undergo extensive safety monitoring prior to its usage in immunization programs. However, no vaccine is entirely without risk. This can be found in lines 60 to 63.

Serious and non-serious should not be confused with severity; based on the outcome of the event (e.g. hospitalisation, death); seriousness is not based on severity; this sentence should be corrected.

 This classification is based on WHO guidelines. The sentence was altered to “Based on the outcome of the event, the World Health Organization (WHO) has classified AEFIs as Serious and Non-serious which is used for regulatory classification”. This can be seen in lines 68 to 70.

Vaccine hesitancy is a public health issue; I would suggest replacing the term “concerns” with vaccine hesitancy:

 The term “concerns” was replaced with “vaccine hesitancy” shown in line 75.

There should be an explanation on why surveillance on immunization is essential; I.e.: to build trust; to reassure the public that AEFIs are being monitored and actions to reduce risks are being taken

 An explanation to why surveillance on immunization safety is crucial has been added to the sentence. This can be seen in line 76 to line 85

AEFI to AEFIs

 AEFI was changed to AEFIs in line 86. 

The term routine is preferable to passive

 The term passive surveillance was changed to routine surveillance in line 87.

HICs also rely on routine systems; under reporting is a general limitation of passive/routine systems; but is there any other reason that LMICs have additional underreporting?

 Poor pharmacovigilance infrastructure was stated as another reason that LMICs have additional underreporting in line 88.

Perhaps rephrase to emphasize the link: underreporting was addressed via the GVSI. One of the targets of… was set.

 The sentence was rephrased as “To address the issue of underreporting the WHO therefore instituted the Global Vaccine Safety Initiative, which set out indicators primarily for monitoring case reporting” for emphasis as seen in lines 88 to 90.

METHODS

Is there a reference for CDC guidelines for evaluating?

 The reference for the CDC guideline has been added to line 136.

How were these guidelines adapted? What were the quantitative and qualitative data that was collected? 

 How the CDC guidelines was adapted has been explained in lines 136 to 138.

Study area paragraph: The evaluation was made through interviewing stakeholders in this region? And reviewing reports in this district? Perhaps clarify what resources in this area was used to do the evaluation.

 The resources in the area used to do the evaluation has been explained in lines 122 to 124.

I don’t think the term detect can be applied here; perhaps just state that caregivers/ vaccinees report to health facilities. How do they report? Using a reporting form?

 The phrase caregivers/vaccinees detect was changed caregivers/vaccinees report adverse events to health facilities. The health facilities record reported events using a standard case reporting form and submits the report to the District Health Directorate. This has been stated in lines 306 to 308

Line 126: What is meant by manage reports? They receive and code them? And they transfer the reports?

 How reports are managed has been explained in lines 308 to 312.

Line 153: title says data collection and analysis, but no analysis was described: perhaps expand on the analysis or remove analysis from title.

 The title “data collection and analysis,” has been changed to “data collection tools and methods”. Nonetheless, Table 1 shows how data collected on attributes was analysed.

Also please make data sources clearer: by stating five different sources of data was collected data included: a) interviews .. b) case-based forms; c) IDSR reports; d) DHIMS2; e) TAC.VBP reports

 From lines 232 to 235 the sources of data were listed as suggested.

Explanation of what TAC-VBP report is?

Line 165: explain what is meant by hypothesis generation; is this signal detection?; 

 TAC-VBP reports explained as Technical Advisory Committee for Vaccines and Biological Products. 

RESULTS

Prefer to use the word understood rather than knew

 The word knew has been changed to understood in line 282.

Section on performance of the surveillance system in meeting objectives: objective were; 1) detect, correct and prevent immunization error-related AEFIs; 2)

identify problems with vaccine lots or brands leading to vaccine reactions, 3) prevent false

blame arising from coincidental adverse events, 4) maintain confidence by properly responding

to community concerns, 5) generate new hypotheses about vaccine reactions and estimate rates

of occurrence of AEFIs.

Use subtitles to indicate the results of assessment of each of these objectives;

 For the section on performance of the surveillance system in meeting objectives, all findings have been grouped under the subtitles which are the 5 objectives of the system. This can be seen in lines 336 to 362.

Would it be better to say 463/476 (97%) of all serious cases were investigated?

 This has been corrected in line 347.

Suggest to rephrase replace the word exonerate; By investigating deaths that followed vaccination and performing causality assessments the system has been able to prevent misinformation speculating that the vaccine attributed to the deaths? Also link this to the objective of prevent false blame arising from coincidental adverse events 

 the phrase the system has been able to exonerate EPI has been revised to the system has been able to prevent misinformation speculating that the vaccine attributed to the death. Seen in line 351.

I would replace the word victim; perhaps individual; 

 The word victim was replaced with individual in line 357

Who paid for the management of the AEFIs?

 Management of the AEFIs is paid for by the EPI, stated in line 357

Does this refer to signal (hypothesis generated)? This needs to be made more clearer

 Hypothesis generation here means to assign possible causes to the reaction. 

how was the usefulness assessed? Was this through interviews? Where did this information come from?

 How usefulness was assessed and the sources of information for this assessment has been explained from lines 364 to 368.

Please do not use the word victim; an individual is suspected to have AEFI; causality is not determined; the word victim gives a negative connotation, when most of the time benefits of the vaccine outweigh the risk

 The word victim was changed to individual in line 

In Ghana is this the case, do caregivers have to go back to the facility to report? Or are there tools in place to allow patient/caregiver reporting?

 As at the time this evaluation was done in January 2019, caregivers have to go back to the facility to report AEFI. Later in 2019, a MedSafety mobile application was developed. This is still being piloted in-country and its implementation is not countrywide.

Does Whatsapp platforms compromise the completeness of reports?

 To prevent issues of incomplete reports, all reports passed through WhatsApp platforms must be accompanied by a case-based form. Nonetheless this gives room for human negligence which can still lead to incomplete report. This was explained in lines 556 and 557

Another limitation is recall bias in interviews; as study was conducted in 2019, investigating from year 2014.

 Study limitation was added in lines 576 to 581.

Proper introduction to and explanation (and referencing) of the tool and how this was applied (e.g. stakeholder interviews and record reviews with reference to the table). I sense that there are both quantitative and qualitative elements in the tool.

 An introduction to and explanation (and referencing) of the tool and how this was applied has been included in lines 135 to 138.

Certain attributes are assessed (presumably part of the evaluation tool) but these are not explicitly related to the objectives of the AEFI surveillance system. 

 Though some attributes are not directly related to the study objectives, they were assessed to understand operations of the system and to explain or discuss findings seen in the attributes of the surveillance system. For instance, knowledge was assessed to understand its relation to underreporting or incomplete reporting,

It may be useful to define the outcome terms in the text when you report the results e.g. PPV of the system detecting the number of vaccine-related AEFI etc.

 Explanations to the outcome terms are in the CDC guidelines referenced in this manuscript. However, PVP has been explained in line 461 as its application is different from that of infectious disease surveillance systems.

Are these results presented? Did you limit the study to AEFI related to EPI only?

 AEFI surveillance in Ghana is limited to EPI. However there were recommendations to the EPI after this evaluation for integration of other units that use vaccinations like the Port Health (for Yellow fever vaccination) and Maternal health Unit (for tetanus vaccination)

General Comments:

Generally, Objectives and measurables need to be made clear; difficult to connect methodology to results: layout needs revision.

Correction

The objective of the study was to assess the attributes of Ghana’s AEFI surveillance system, its usefulness and to determine whether the system is meeting its objectives. This was stated in the last sentence of the introduction. However, to connect these objectives to the methods and results, sub-headings (for each objective) have been introduced in the method and results section to improve consistency.

Also, the layout of the report has been revised for clarity. The operations of the system has been moved to the results section. For the assessment of system meeting its objectives, all findings have been grouped under the subtitles which are the 5 objectives of the system. 

Additional requirements:

 1.) Please ensure that your manuscript meets PLOS ONE's style requirements, including those for file naming. The PLOS ONE style templates can be found at https://journals.plos.org/plosone/s/file?id=wjVg/PLOSOne_formatting_sample_main_body.pdf

 and

Response 

The format of the manuscript is revised according to the PLOSE ONE style

Thank you for your comments and inputs

Sincerely,

Joseph Asamoah Frimpong

---

## [Editor Report · Decision Letter 1]

22 Dec 2021

PONE-D-21-21870R1Evaluation of the Adverse Events Following Immunization Surveillance System, Ghana, 2019PLOS ONE

Dear Dr. Frimpong,

Thank you for submitting your manuscript to PLOS ONE. After careful consideration, we feel that it has merit but does not fully meet PLOS ONE’s publication criteria as it currently stands. Therefore, we invite you to submit a revised version of the manuscript that addresses the points raised during the review process.

Many thanks and apologies for the delay. The presentation is much clearer. Lines 229-270, the case definitions, should be within the Methods Section.

We look forward to receiving your revised manuscript.

Kind regards,

Emma K. Kalk

Academic Editor

PLOS ONE
---

## [Author Response · Author response to Decision Letter 1]

7 Feb 2022

The AEFI case definitions, have been put under data collection within the Methods Section. This can be seen in lines 132 to 142.

---

## [Editor Report · Decision Letter 2]

16 Feb 2022

Evaluation of the Adverse Events Following Immunization Surveillance System, Ghana, 2019

PONE-D-21-21870R2

Dear Dr. Frimpong,

We’re pleased to inform you that your manuscript has been judged scientifically suitable for publication and will be formally accepted for publication once it meets all outstanding technical requirements.

Kind regards,

Emma K. Kalk

Academic Editor

PLOS ONE
---

## [Editor Report · Acceptance letter]

21 Feb 2022

PONE-D-21-21870R2 

Evaluation of the Adverse Events Following Immunization Surveillance System, Ghana, 2019. 

Dear Dr. Frimpong:

I'm pleased to inform you that your manuscript has been deemed suitable for publication in PLOS ONE. Congratulations! Your manuscript is now with our production department. 

Kind regards, 

on behalf of

Dr. Emma K. Kalk 

Academic Editor

PLOS ONE